# Unsupervised Discovery and Composition of Object Light Fields

## Abstract

Neural scene representations, both continuous and discrete, have recently emerged as a powerful new paradigm for 3D scene understanding. Recent efforts have tackled unsupervised discovery of object-centric neural scene representations. However, the high cost of ray-marching, exacerbated by the fact that each object representation has to be ray-marched separately, leads to insufficiently sampled radiance fields and thus, noisy renderings, poor framerates, and high memory and time complexity during training and rendering. Here, we propose to represent objects in an object-centric, compositional scene representation as light fields. We propose a novel light field compositor module that enables reconstructing the global light field from a set of object-centric light fields. Dubbed Compositional Object Light Fields (COLF), our method enables unsupervised learning of object-centric neural scene representations, state-of-the-art reconstruction and novel view synthesis performance on standard datasets, and rendering and training speeds at orders of magnitude faster than existing 3D approaches.

## 1 Introduction

A critical aspect of scene understanding is parsing the scene into its composite parts. On the highest level, these are the static scene elements as well as rigid objects. It is this scene decomposition that enables us to quickly understand and subsequently interact with our environment, and it is relevant to downstream tasks ranging from robotics to autonomous navigation to generative modeling.

A recent exciting avenue of research leverages generative modeling to learn scene decomposition in an unsupervised manner. One line of work accomplishes this by modeling objects in 2D images (Eslami et al., 2016; Crawford & Pineau, 2019; Kosiorek et al., 2018; Lin et al., 2020; Jiang et al., 2019; Burgess et al., 2019; Greff et al., 2019; 2016; 2017; Engelcke et al., 2019; Locatello et al., 2020). However, these models lack a critical aspect of scene understanding, which is the reconstruction of the underlying 3D structure. A recent line of work has instead proposed to model objects directly in 3D via object-centric neural scene representations (Yu et al., 2021; Stelzner et al., 2021). Leveraging differentiable rendering, these models can be trained just from 2D image observations, offering exciting new perspectives on the unsupervised learning of visual representations useful to downstream tasks such as robotics, tracking, and scene understanding.

While promising, a fundamental limitation of these models is that they suffer from the high computational complexity of volume rendering. Rendering images from a *single* neural radiance field is computationally costly as it requires dense sampling of 3D points along each ray (Mildenhall et al., 2020). Rendering of object-centric representations is more expensive yet, as each object radiance field needs to be ray-marched separately and rendering cost thus scales with the number of objects in the scene. This high computational complexity prevents existing methods from achieving real-time rendering, and further limits the number of objects in each scene to about 10 objects. It further limits the number of samples per ray, leading to low-quality renderings due to insufficiently sampled object radiance fields. Though recent progress has been made in speeding up volume rendering in the case of reconstructing a single scene, applying similar techniques to the domain of generalization and prior-based reconstruction is an open problem.

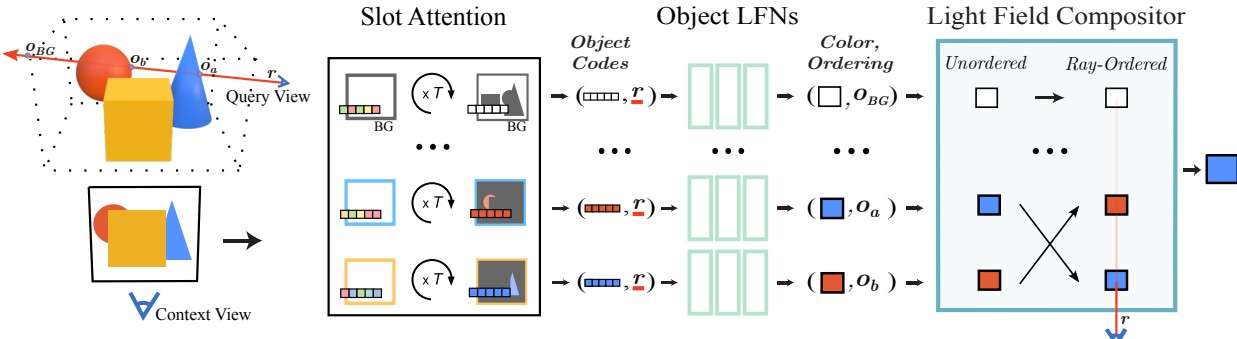

Figure 1: **Overview.** We represent scenes as Compositional Object Light Fields (COLF). Given an image, we first use slot attention to infer a set of object codes describing detected objects in the scene. Each object code is queried for a color, as well as an ordering value (related to the camera depth) observed at the first intersection of the object and a novel ray $r$. Our Light Field Compositor module then determines ray-visibility weights and yields the color observed by $r$.

In this work, we overcome this limitation. We propose modeling a scene not as a composition of 3D elements, but as a composition of *neural object light fields.* Rendering a neural object light field only requires sampling the neural representation exactly once per ray. However, in contrast to volume rendering, compositing of object-centric light fields is not analytical, as their per-ray depth order is unknown. Thus, we propose a novel light field compositor module that learns to estimate soft visibility for each object ray query, enabling us to compose a set of object-centric light fields into the light field of the complete scene. We demonstrate that our method not only enables critical speed-up and memory reduction, but also outperforms previous state-of-the-art methods in terms of reconstruction quality, and enables editing to compose and render unbounded scenes with tens of objects.

In summary, we make the following contributions:

- We propose a novel method that leverages neural light fields for the unsupervised discovery of objects.

- We address the challenge of non-analytical compositing of partial light fields by proposing a parametric light field compositing module.

- Our method significantly reduces the memory requirement and time complexity, enabling real-time rendering at state-of-the-art reconstruction quality.

## 2    Related Work

**Neural Scene Representation and Rendering:** Our method is related to recent work on inferring latent parameters of 3D scenes from images. Eslami et al. (2018) proposed to encode several image observations of a scene into a latent code that can subsequently be decoded into novel views with a convolutional neural network. 3D voxelgrids, combined with differentiable ray-marching, first allowed self-supervised discovery of shape and appearance from images (Sitzmann et al., 2019a; Lombardi et al., 2019). Inspired by neural implicit shape representations (Park et al., 2019; Mescheder et al., 2019; Chen & Zhang, 2019), neural-field based representations, combined with neural rendering, lifted limitations of resolution (Sitzmann et al., 2019b; Niemeyer et al., 2020; Mildenhall et al., 2020; Yariv et al., 2020; Xie et al., 2021). By conditioning on latent variables, this enables 3D reconstruction from just a single observation (Sitzmann et al., 2019b; Niemeyer et al., 2020). By conditioning locally on image features (Saito et al., 2019) and constraining ourselves to learning about local patches instead of object-centric or scene-centric representations, we may generalize to scenes with unseen numbers of objects (Yu et al., 2020; Trevithick & Yang, 2020). Most recently, light fields have been proposed as an alternative to 3D-structured representations to address the extraordinary cost of rendering (Sitzmann et al., 2021): Instead of mapping a 3D coordinate to whatever exists at that coordinate

and thus requiring ray-marching for rendering, they directly map an oriented ray to whatever is observed by that oriented ray, therefore rendering with a *single* evaluation of the neural network per ray. However, these approaches do not infer object-centric representations, lacking semantic 3D scene understanding.

**2D Compositional Scene Representations:** Deep-learning based inference of object-centric representations was first addressed by factorizing 2D images into 2D components, either represented as localized object-centric patches (Eslami et al., 2016; Crawford & Pineau, 2019; Kosiorek et al., 2018; Lin et al., 2020; Jiang et al., 2019) or scene mixture components (Burgess et al., 2019; Greff et al., 2019; 2016; 2017; Engelcke et al., 2019). Locatello et al. (2020) proposed Slot Attention as an inference model for such object-centric representations, which we also rely on in the present work. Though slot attention has in the past been limited to toy scenes, recent, concurrent work has demonstrated extensions to more complex scenarios. Improvements include leveraging weak supervision in the form of optical flow and object bounding-boxes (Kipf et al., 2021), geometry supervision in the form of depth maps (Elsayed et al., 2022), or attention-based instead of mixture-based decoding (Singh et al., 2022). Our work is orthogonal: we do not propose any improvements to the inference algorithm, but rather, propose an alternative object rerpresentation in the form of 3D object light fields, enabling 3D object-centric representation learning. Improvements in inference as discussed in this concurrent work might thus also benefit the proposed method.

**3D Compositional Scene Representations:** Recent work has addressed unsupervised 3D scene decomposition. Elich et al. (2020) infer object shapes from a single scene image, but require ground-truth shapes for pre-training. Chen et al. (2020) extend the Generative Query Network (Eslami et al., 2018) to decompose 3D scenes, but require multi-view observations during inference. Bear et al. (2020) model a scene as a scene graph and infer parametric object-centric shapes. Block-GAN and GIRAFFE (Nguyen-Phuoc et al., 2020; Niemeyer & Geiger, 2020) build unconditional generative models for compositions of 3D-structured representations, but only tackle generation, not reconstruction. Closest to our method are Stelzner et al. (2021) and Yu et al. (2021), who use slot-based encoders for unsupervised discovery of objects. However, both approaches use volumetric neural scene representations (Mildenhall et al., 2020) - due to the severe cost of rendering at both training and test time, these approaches cannot render at real-time frame-rates and require either single-digit batch-sizes (Yu et al., 2021) or ground-truth depth to accelerate training (Stelzner et al., 2021), and even then suffer from rendering artifacts due to insufficient volumetric sampling. While recent work has achieved impressive progress in accelerating the reconstruction and rendering of radiance fields for *single* scenes (Müller et al., 2022; Fridovich-Keil and Yu et al., 2022; Garbin et al., 2021; Neff et al., 2021), applying these principles to the regime of prior-based reconstruction from few observations is an open problem. That is, because these methods leverage sparsity-based techniques, such as skipping empty space or using sparse data structures, directly inferring the scene structure in a feedforward inference setting is difficult or at least has not yet been demonstrated. Our method not only significantly outperforms prior object-centric approaches in terms of reconstruction quality, but also addresses the computation and memory complexity of volumetric rendering.

**Layered Representations for View Synthesis:** Prior works leverage the multiplane image scene representation, a set of fronto-parallel RGBA planes predicted at various depth values, for the task of novel view synthesis (Zhou et al., 2018; Srinivasan et al., 2019). While this line of work is similar in the spirit of combining scene decomposition and novel view synthesis, our decomposition is semantic (into objects), whereas their decomposition is geometric (into depth planes), and our view synthesis supports full 360-degree scenes, whereas theirs only supports front-facing scenes.

**Light Field Compositing:** Some prior work has investigated the composition of light fields parameterized as multi-plane images (Mildenhall et al., 2019; DuVall et al., 2019). Chen et al. (2006) generalize the image alpha-compositing operator to light fields. However, both of these techniques are incompatible with object-centric light fields, as they assume front-facing (i.e. not 360-degree) scenes, and, critically, that the *depth order* along a ray is known. Alas, this is not the case when a scene is described by a set of object-centric light field networks that can be rendered from arbitrary 360-degree perspectives, which we address by introducing the light field compositor module.

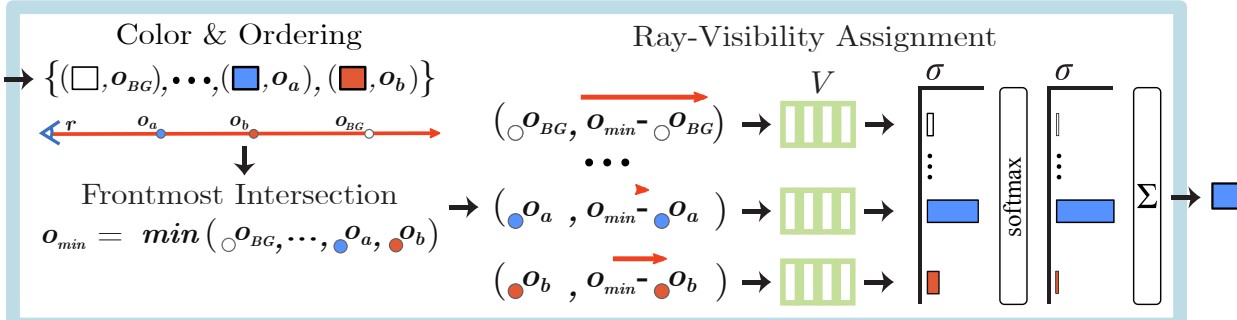

Figure 2: **Light Field Compositor.** Our light field compositor reasons about the relative order of surface-ray intersections for each of the object light fields, producing a set of softmax-weighted visibility scores. We then composite the contributions from each object light field into a single color while respecting occlusions. See Sec. 3.2 for details.

## 3 Method

Our goal is to build and train a model that, given a single image of a 3D scene, can parse it into a set of $K$ object-centric, 3D-aware representations and enables us to re-render from novel views. In Sec. 3.1, we review the light field networks (Sitzmann et al., 2021) neural scene representation. In Sec. 3.2, we introduce our new light field compositor that enables rendering of a 3D scene from novel view points via occlusion-aware compositing of $K$ light field networks. In Sec. 3.3, we describe our new end-to-end auto-encoder model, the encoder, the losses, and the training strategy that enables us to learn to decompose scenes into object light fields. Fig. 1 gives an overview over the proposed model.

### 3.1 Light Field Networks (LFNs)

Recently, Sitzmann et al. (2021) proposed to represent a 3D scene by directly parameterizing its 360-degree *light field* (Levoy & Hanrahan, 1996; Gortler et al., 1996; Adelson et al., 1991) via a neural network. A scene is then parameterized as an MLP $\Phi$ that maps every 6-dimensional oriented ray $\mathbf{r}$ directly to the color observed by that ray: $\Phi(\mathbf{r}) = \mathbf{c}$. Instead of sampling $\Phi$ hundreds or thousands of times as in 3D-structured scene representations, rendering in light field networks reduces to a *single* sample of $\Phi$ per pixel, achieving a dramatic speed-up and reduction in memory complexity of rendering. Since $\Phi$, as an MLP, is differentiable, it can be trained by minimizing the L2 loss between the rendered colors $\Phi(\mathbf{r})$ and the colors of the training image $\mathcal{I}(\mathbf{r})$, i.e., $\mathcal{L} = \|\Phi(\mathbf{r}) - \mathcal{I}(\mathbf{r})\|_2^2$. We may also learn a prior over light fields that enables reconstruction from only a *single* observation by generalizing over a set of 3D scenes, where each 3D scene is represented by a latent code $\mathbf{z}$, and conditioning the light field network on that latent code, which we denote as $\Phi(\mathbf{r}; \mathbf{z})$ (Sitzmann et al., 2021).

### 3.2 Compositing LFNs

We wish to decompose 3D scenes into their object parts by representing each object via a separate Light Field Network (LFN). A scene is then represented as a set of $K$ LFNs $\Phi_1, \Phi_2, ..., \Phi_K$. To formulate a differentiable rendering algorithm, we require a function that maps from a given ray $\mathbf{r}$ to the color $\mathbf{c}$ observed by that ray. We consider scenes without transparency, where the color of a given ray is determined only by the first surface it intersects, and therefore, only by one of the $K$ LFNs. We must thus formulate an algorithm that composites the set of LFNs according to their relative depth ordering, such that we yield the color $\mathbf{c}_j$ of the object with index $j$ exactly if that object was hit by the ray $\mathbf{r}$ *first*, occluding the color obtained from other LFNs.

In the case of neural radiance fields (Yu et al., 2021; Stelzner et al., 2021), composition is an analytical, non-parametric computation. This is because volume rendering samples *points along a ray*, and therefore is aware of the *order* of the densities along the ray. This allows an analytical alpha-compositing rendering function even in the case of compositing several object-centric representations to a single scene representation

(Yu et al., 2021). At first glance, an equivalent approach in the compositing of light fields might have each object-centric light field map a ray to a color $\mathbf{c}_i$ and an alpha value $\alpha_i$, and then alpha-composite the colors to obtain the final color of the ray, as previously proposed in Chen et al. (2006). However, this assumes that the order of the object light fields along the ray $\mathbf{r}$ is known. This is not the case: sampling an LFN does not yield depth, and thus, the relative order of the surfaces observed in each of the LFNs is unknown. While it is possible to compute depth via the derivatives of the light field (Sitzmann et al., 2021), this depth is sparse, and thus cannot be used for compositing at every pixel.

We thus propose a Light Field Compositor model - please see Fig. 2 for an overview. First, we extend LFNs to encode additional information in each of the object light fields $\Phi_i$ that allows us to compute their relative ordering. Specifically, we now map each ray $\mathbf{r}$ to a tuple of color $\mathbf{c}_i$ and *ordering value* $o_i$:

$$\Phi_i : \mathbb{R}^6 \to \mathbb{R}^3 \times \mathbb{R}^1; \quad \Phi_i(\mathbf{r}) = (\mathbf{c}_i, o_i) \tag{1}$$

We note that the ordering value is not equivalent to camera depth in the conventional sense. This is because the output of the LFN is invariant to the query camera's position along the query ray (Sitzmann et al., 2021). Therefore, the standard notion of regressing a depth value which increases as the query camera moves further away from the scene intersection is not possible, as the output of the LFN by design does not change as the query camera moves along that ray. Rather, the ordering value can intuitively be understood as the signed distance of each light field's ray-intersection relative to the projection of a global reference point, such as the world origin or the context camera position, onto the query ray $\mathbf{r}$.

Given a set of $K$ light field networks and a ray $\mathbf{r}$, we query each LFN to obtain color and ordering values $(c_i, o_i) = \Phi_i(\mathbf{r})$. We then use the minimum of the ordering values $o_{\min} = \min(o_1, ..., o_k)$ to map the $i$'th light field's ordering value to a visibility value $v_i = V(o_i, o_{\min} - o_i)$, where $V$ is a small MLP. The difference $o_{\min} - o_i$ intuitively corresponds to how far behind the light field $i$'s intersection is from the first estimated intersection. The visibility scores are then softmax-normalized and used to determine the color contribution from each light field, allowing us to obtain the final ray color as a weighted sum of the per-slot radiances:

$$\mathcal{I}'(\mathbf{r}) = \Sigma_{i=1}^K \mathbf{c}_i \frac{\exp(v_i)}{\Sigma_{j=1}^K \exp(v_j)} \tag{2}$$

We considered explicitly making the ordering values relative to the projection of the context camera position onto the query ray, achieved by subtracting from each ordering value the distance from the camera's projection point to the query ray origin. However, because such a shifting would preserve the relative ordering-value differences between objects, and given that the visibility scores are softmax-normalized, the projection-based shifting is unnecessary.

Our compositor is related to prior work on differentiable mesh renderers, such as (Liu et al., 2019; Kato et al., 2018), which use a handcrafted distance kernel to assign higher visibility to the frontmost ray-intersecting face than subsequent faces along the ray. These handcrafted kernels, however, are tuned for the depth range and amount of gradient support needed by the application (Liu et al., 2019). We instead employ a learned weighting kernel, implemented by the small network $V$. This learned weighting kernel allows the network to adapt to the depth range of each dataset, whereas a handcrafted kernel requires fine-tuning for these two considerations.

We visualize the predicted raw and minimum-subtracted ordering values on our city-block dataset in Fig. 4. Note how in the minimum-subtracted ordering value plot, the subject of each slot has a value of zero and all other subjects have a value below zero, yielding an easier task for the occlusion kernel.

### 3.3 Learning to Represent Scenes as Sets of LFNs

We now describe a full encoder-decoder architecture that will enable us to learn, in an unsupervised manner, a model capable of decomposing a scene into a set of object-centric Light Field Networks (LFNs) given a *single* image. An overview can be seen in Fig. 1. The encoder module will map a context image $\mathcal{I}_{ctxt}$ to a set of $K$ latent codes $\mathbf{z}_1, \mathbf{z}_2, ..., \mathbf{z}_K$. Each of these latent codes will be used to condition an object-centric light

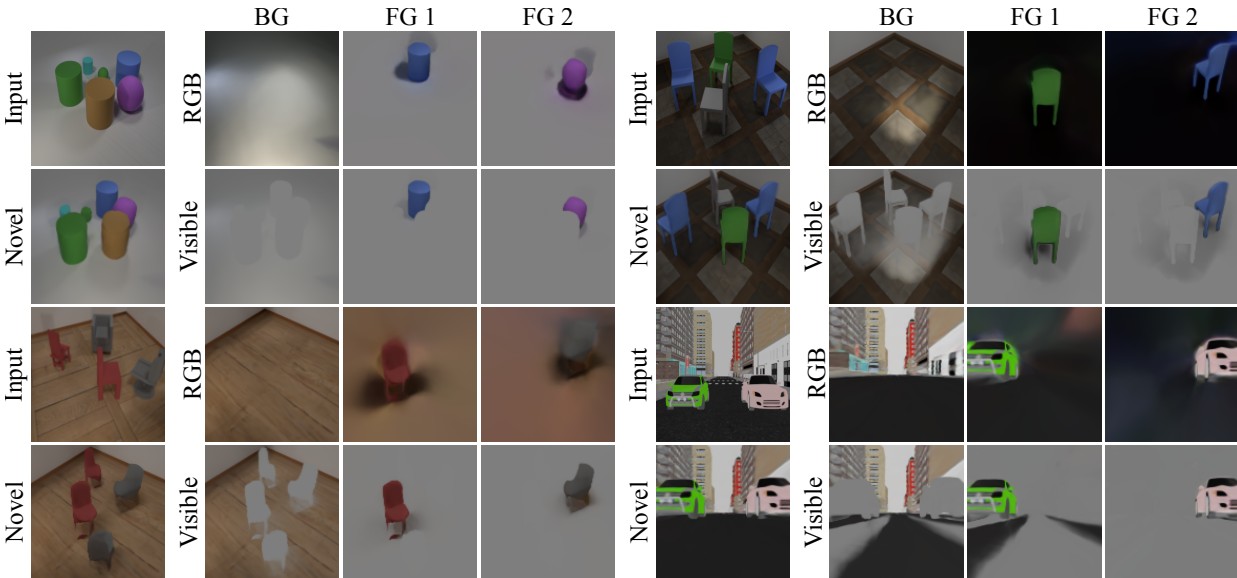

Figure 3: **Qualitative Scene Decomposition Results.** We show light field decompositions on four scenes. For each scene, the leftmost column shows the model input (top) and one novel (bottom) view. To the right of each scene, the first row shows the RGB prediction from a set of slots, and the bottom row shows the corresponding visibility mask, which filter out occluded rays prior to their composition into the final image. Please see the supplemental material for video results.

field network $\Phi_i(\mathbf{r}) = \Phi(\mathbf{r}; \mathbf{z}_i)$, defined in the coordinate frame of the context image $\mathcal{I}_{ctxt}$. Following (Yu et al., 2021), we further introduce a *background light field* $\Phi_{BG}(\mathbf{r}) = \Phi(\mathbf{r}; \mathbf{z}_{BG})$, whose latent is inferred by a separate encoder, and which is defined in a canonical world coordinate frame. Given a *query* image $\mathcal{I}_{query}$ of the same 3D scene, we parameterize the per-pixel camera rays via the explicit and implicit camera parameters, and sample each $\Phi_i(\mathbf{r})$ and $\Phi_{BG}(\mathbf{r})$ to yield a set of colors and ordering values $(\mathbf{c}_1, o_1), (\mathbf{c}_2, o_2), ..., (\mathbf{c}_K, o_K)$ and background values $(\mathbf{c}_{BG}, o_{BG})$. Finally, we leverage our compositor module to render a color for each ray, compute a reconstruction loss, and backpropagate that loss to train our model end-to-end.

**Encoder:** Our encoder maps an image to the set of latent vectors $\{\mathbf{z}_i\}_{i=1}^{K}$ parameterizing the object-centric light field networks that exist in the scene. We first concatenate fixed pixel coordinates to the image color channels and encode the image into a feature map using a U-net (Ronneberger et al., 2015) encoder. We follow the convention of uORF (Yu et al., 2021) and concatenate constant pixel coordinates without camera information, opposed to how (Stelzner et al., 2021) adds camera position and ray direction channels, to describe the objects in camera space rather than world space, which has shown to generalize more robustly to novel viewpoints and objects (Tatarchenko et al., 2019). To map the feature grid to a set of latent vectors describing the objects in the image, we use the Slot Attention module (Locatello et al., 2020), which achieves this by sampling a set of latent vectors, or "slots", from a learned slot distribution, and having them dynamically specialize to image entities via recurrent competition to explain parts of the image. The previous work uORF (Yu et al., 2021) leverages the observation that the geometry and appearance of background and foreground elements in images are significantly different and that the model could benefit from disentangling their latent spaces by using a separate slot distribution for each. This modification of sampling one slot from a background slot distribution and the remaining slots from a foreground slot distribution enabled their work to be the first to reconstruct scenes with textured backgrounds of significant complexity; we adopt this architectural change as well. The pseudo-code describing the background-aware slot encoding is the same as in uORF, but exists in the supplemental material for reference.

**Background Light Field:** Following uORF (Yu et al., 2021), we represent the background via a conditional light field network $\Phi_{BG}(\mathbf{r}) = \Phi(\mathbf{r}; \mathbf{z}_{BG})$. $\Phi_{BG}(\mathbf{r})$ is defined in *world* space, while all foreground scene components $\Phi_i$ are defined in camera space. The intuition for this design choice is that since the background

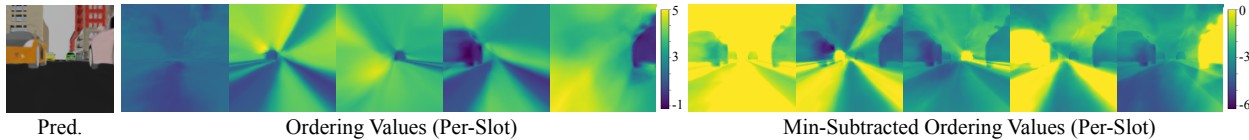

Pred.                    Ordering Values (Per-Slot)                    Min-Subtracted Ordering Values (Per-Slot)

Figure 4: Predicted ordering values (per-slot) for a city-block scene and its minimum-subtracted comparison.

geometry is always canonicalized in world space, it is easier for the model to learn the background geometry and appearance in world space, whereas since the foreground elements are randomly rigidly transformed in world space, it is easier for the foreground components to be queried in camera space rather than have the model perform an implicit camera to world space transformation.

**Losses:** Our model encodes an input image and renders novel views $\mathcal{I}'$ of the underlying scene at a set of query camera positions. We supervise the model with the L2 reconstruction loss $\|\mathcal{I}_{query} - \mathcal{I}'\|^2$, where $\mathcal{I}_{query}$ are the ground truth views. We use a deep-feature based perceptual loss (Zhang et al., 2018) on both chair datasets to avoid inherent ambiguities in estimating lighting and geometry at occluded views. Lastly, we impose a small penalty $\|z\|^2$ on each object regressed code $z$ to enforce a Gaussian prior.

**Curriculum Schedule:** Empirically, slot encoders are limited in the complexity of scenes that they can disentangle into parts, leading to a prevalance of scenes consisting of simple geometric shapes of uniform color (Eslami et al., 2018; Burgess et al., 2019; Lin et al., 2020). This can by explained by two factors. One is how the difficulty of learning additional components which define a factorized slot representation increases with the object complexity and variance. These factorized scene components include the slot distribution parameters to accurately describe the space of objects, decoder weights to reconstruct the factorized objects, and the compositing of the decoded objects into the original domain. And second, there is no principled reason for a factorized scene representation to emerge other than compression, i.e., the difficulty of encoding a complex scene with a monolithic description, and encourage the model to escape this local minima. We observe that when the state-of-the-art model (Yu et al., 2021) is evaluated on a dataset of *textured* chairs, rather than the uniformly colored chairs they report success on, decomposition fails to emerge as a result of the increased object complexity complicating some combination of the first two factorized-representation components discussed. This shortcoming is exacerbated in our model, which has fewer constraints on multi-view consistence as 3D-structured representations and has to learn additional parameters of a neural compositor, due to the nature of neural light fields (Sitzmann et al., 2021). We observe that our model similarly has high variance in learning factorized representations on a dataset of uniformly colored chairs. However, similar to prior work, our model consistently and reliably learns a factorized representation on the CLEVR-567 dataset, due to the decreased object complexity. We thus propose a curriculum learning strategy, where we use the CLEVR-567 to learn a prior of a factorized representation before training on more complex scenes. We find that this factorized prior is strong and leads to immediate convergence of a factorized representation on each subsequent dataset evaluation. This strategy, given the datasets we evaluate on (see Sec. 4.1), is to initialize the models for Room-Chair, Room-Diverse, and City-Block with the weights of a model that has been trained on the CLEVR-567 dataset. We note that this incremental learning curriculum is likely a viable strategy to advance the scene complexity bound of slot-based auto-encoders in general. Lastly, we initially render and supervise images at $64 \times 64$ resolution to efficiently learn the coarse structure and decomposition of scenes, and subsequently supervise at $128 \times 128$ to learn more fine object structure. Note however, that we do not need to employ higher-resolution supervision via cropped images, as the baseline (Yu et al., 2021) does, thanks to the efficiency of the light field decoder.

## 4 Experiments

We demonstrate that compositional neural light fields, unconstrained by the sampling requirements of volumetric rendering, outperform prior work on unsupervised learning of object-centric 3D representations while dramatically reducing time and memory complexity. We further demonstrate that object-centric light fields admit scene editing in the from of translation and composing, and allow rendering of scenes with tens of

(a) Decomposition Results

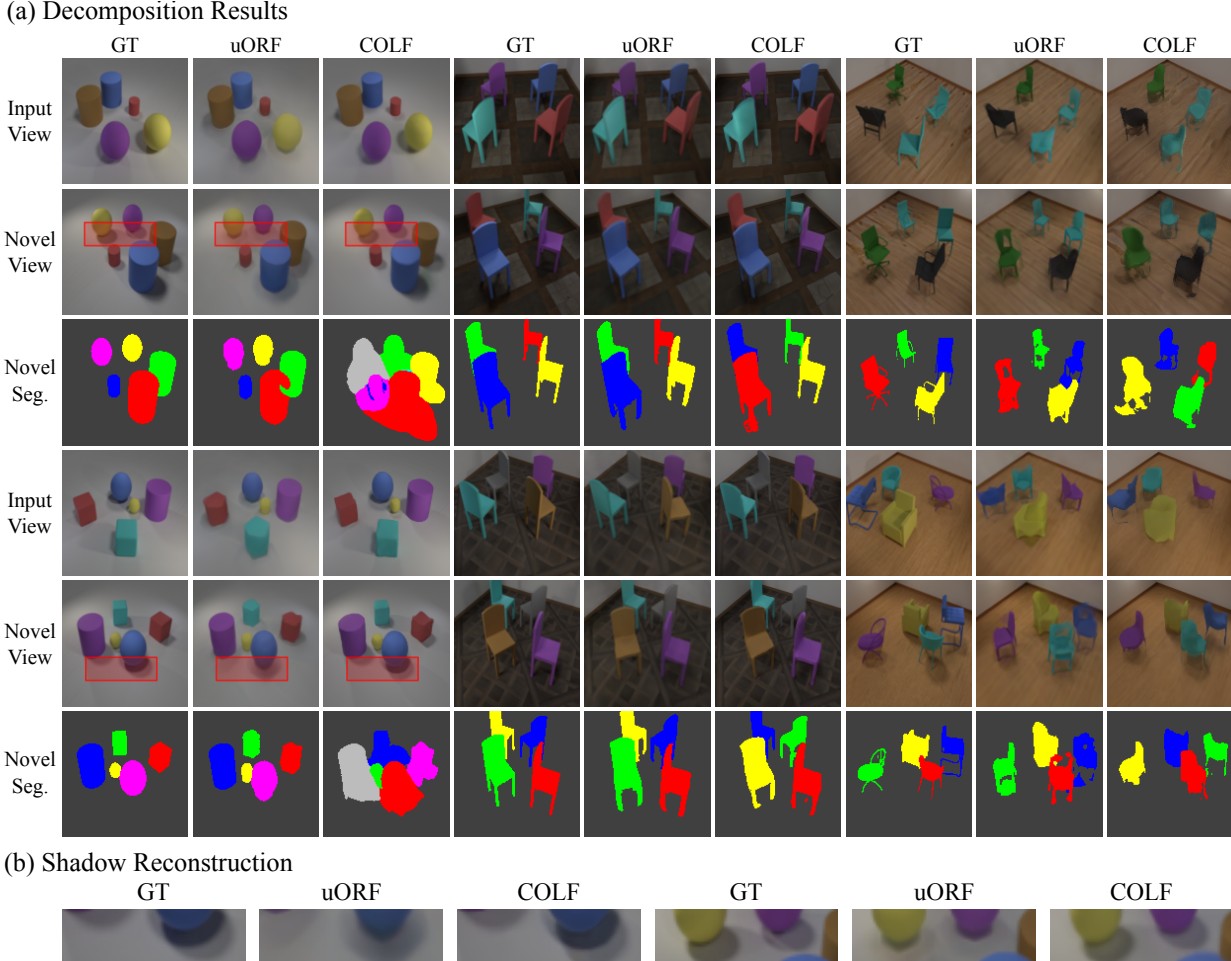

(b) Shadow Reconstruction

Figure 5: **Qualitative Comparison.** In (a), from top to bottom, we compare reconstructions of the input view, a novel view, as well as the novel view segementation by the state-of-the-art baseline uORF (Yu et al., 2021) and our method, across CLEVR, chairs-simple, and chairs-diverse. We achieve higher reconstruction quality across each dataset while being orders of magnitude more efficient. Interestingly, our method (arguably correctly) assigns shadows to their object slots rather than the background slot and reconstructs them significantly better, as highlighted in (b). For video results, please see the supplement.

Table 1: **Quantitative Comparison.** Our method outperforms state-of-the-art baselines (Yu et al., 2021) across all reconstruction quality metrics, while being orders of magnitude faster and requiring less memory. We also find that COLF often captures shadows where uORF does not.

| Model | CLEVR-567 | | | Room-Chair | | | Room-Diverse | | |
|---|---|---|---|---|---|---|---|---|---|
| | LPIPS ↓ | SSIM ↑ | PSNR ↑ | LPIPS ↓ | SSIM ↑ | PSNR ↑ | LPIPS ↓ | SSIM ↑ | PSNR ↑ |
| NeRF-AE (Yu et al., 2021) | 0.1288 | 0.8658 | 27.16 | 0.1166 | 0.8265 | 28.13 | 0.2458 | 0.6688 | 24.80 |
| uORF (Yu et al., 2021) | 0.0859 | 0.8971 | 29.28 | 0.0821 | 0.8722 | 29.60 | 0.1729 | 0.7094 | 25.96 |
| COLF (ours) | **0.0608** | **0.9346** | **31.81** | **0.0485** | **0.8934** | **30.93** | **0.1274** | **0.7308** | **26.02** |

objects at interactive frame-rates. Please find further qualitative results, including video, in the supplemental material.

Table 2: **Quantitative Segmentation metrics.** On segmentation metrics of room-scenes, our method performs approximately on par with uORF while being orders of magnitude faster and achieving better reconstruction quality. Note that on CLEVR, COLF performs worse because it more accurately reconstructs shadows, assigning those "background" pixels to the foreground; this is reflected in the FG-ARI comparison

| | CLEVR-567 | | | Room-Chair | | | Room-Diverse | | |
|---|---|---|---|---|---|---|---|---|---|
| Model | ARI ↑ | NV-ARI ↑ | FG-ARI ↑ | ARI ↑ | NV-ARI ↑ | FG-ARI ↑ | ARI ↑ | NV-ARI ↑ | FG-ARI ↑ |
| Slot Attention (Locatello et al., 2020) | 3.5 | - | **93.2** | 38.4 | - | 40.2 | 17.4 | - | 43.8 |
| uORF (Yu et al., 2021) | **86.3** | **83.8** | 87.4 | 78.8 | 74.3 | 88.8 | 65.6 | **56.9** | 67.9 |
| COLF (ours) | 59.5 | 46.6 | 92.6 | **83.9** | **83.5** | **92.4** | **70.7** | 54.5 | **71.7** |

## 4.1 Setup

**Baselines.** On both tasks of scene decomposition and novel view synthesis, we compare to the state-of-the-art method uORF (Yu et al., 2021). Since the proposed method and uORF are the only two models in this unsupervised 3D-aware object discovery regime, we also provide a reference point to models without these inductive biases — the 2D state-of-the-art Slot Attention model (no 3D inductive bias) (Locatello et al., 2020) on the task of scene decomposition, and a NeRF (Mildenhall et al., 2020) autoencoder without any compositionality inductive bias dubbed "NeRF-AE". Note that the recently proposed GIRAFFE (Niemeyer & Geiger, 2020) is an unconditional generative model and can thus not serve as a competitive baseline, as it cannot reconstruct a multi-object scene from a given image (Yu et al., 2021).

**Datasets.** We use four datasets. First, to compare our model with uORF on the tasks of scene segmentation and novel view synthesis, we evaluate models on their proposed three room-scene datasets of increasing scene complexity. Further, to demonstrate compositing of multiple scenes and rendering of resulting scenes with many objects, we also introduce a new synthetic dataset of a long city block scene with two lanes of car traffic.

**CLEVR-657:** The first room-scene dataset proposed by (Yu et al., 2021) is a 3D extension to the CLEVR (Johnson et al., 2017) dataset. A textureless room is populated with five to seven simple geometric shapes (cubes, cylinders, and spheres) and "Rubber" material without specularity. There are 1,000 scenes for training and 500 for testing.

**Room-Chair:** The second room-scene dataset is populated with three to four chairs of the same geometry, and the room features three different floor textures. There are 1,000 scenes for training and 500 for testing. Views of each scene are captured with a camera at fixed elevation and randomly sampled azimuth, pointed at the room center.

**Room-Diverse:** The third room-scene dataset is populated with three to four chairs of the geometry sampled from 1,200 ShapeNet chairs (Chang et al., 2015), and the room features fifty different floor textures. There are 5,000 scenes for training and 500 for testing. Views of each scene are captured with a camera at fixed elevation and randomly sampled azimuth, pointed at the room center.

**City-Block:** To demonstrate editing and composition of scenes with many objects and large depth range, we place instances of four car geometries from the ShapeNet (Chang et al., 2015) dataset on a two-lane city block. A natural difficulty here is to provide the model sufficient context of all cars placed through the city block at once. To address this, we offer several context views per scene, captured in front of each row of cars placed. At train time, we populate the scene with two rows of cars (two context views). The camera is always facing forwards and varies in height from the car height to just above the ground and depth-wise from the beginning to end of the city block. There are 500 scenes for training and we render out one scene for qualitative demonstration.

**Implementation Details.** On CLEVR-567, we set the background latent vector to 0, since the model needs no information about the unchanging background. This leads to faster model convergence. On City-Block,

we discard the background latent and instead use a dedicated background light field network. We pretrain this network on all training images.

## 4.2 Unsupervised Scene Decomposition

**Setup:** For each test scene in all three room-scene datasets, we encode one of the scene's four captured images as the input view and use the remaining three for evaluation on novel views. We render novel views at the camera positions of the three ground truth query views, and infer segmentation masks for evaluation from the compositor module's slot masks. Specifically, we map a pixel $p$ in the rendered view to one of the model's slots by assigning it the slot to which the compositor has yielded the highest contribution weight at $p$.

**Metrics:** We use the Adjusted Rand Index (ARI) metric to evaluate our inferred segmentation masks against ground truth segmentation masks. To quantitatively compare with the baseline (Yu et al., 2021), we evaluate with three versions of the ARI: (1) ARI on only the input view , (2) ARI on only the three novel views (ARI-NV), and (3) ARI on only the foreground elements (ARI-FG).

**Results:** We report quantitative comparisons to the baseline (Yu et al., 2021) in Tbl. 2 and illustrate qualitative comparisons in Fig. 4. The results in Tbl. 2 show that our architecture performs well on the task of unsupervised scene decomposition — often outperforming the baseline (Yu et al., 2021) and is competitive when otherwise. The glaring exceptions are the FG-ARI and NV-ARI scores on the CLEVR-567 dataset, where we benchmark significantly worse. However, as can be seen in the qualitative comparison on the CLEVR-567 scene in Fig. 4, our model performs "worse" when compared to the ground truth segmentation results because our model *better* reconstructs the shadows of foreground objects than (Yu et al., 2021), which is penalized since the ground truth object masks do not include their shadows. Although some may regard this behavior of assigning shadows to the object inducing them as "incorrect" segmentation, particularly when compared to typical segmentation datasets where shadows are manually assigned to the background, note there is no principled way for the model to decide whether shadows are assigned to the background or foreground unless we hand-craft a prior. Especially when considering that the only training signal is novel view synthesis, we argue our model's behavior is *more* correct, as the rendered shadows are causally related to the objects — if the object inducing the shadow was removed, the shadow should be removed as well. Thus we want to stress this low score is *not* evidence that our model cannot form disentangled representations. In fact, we outperform the baseline (Yu et al., 2021) considerably when only considering foreground pixels, confirming that despite this questionable metric, our model indeed forms factorized representations which segment objects in the scene well.

## 4.3 Novel View Synthesis

**Setup:** Similar to the scene decomposition setup, for each test scene in all three room-scene datasets, we reserve one view as an input view and use the remaining three to evaluate the novel view reconstructions. We render novel views at the query camera positions and compare the reconstructed images with the metrics listed below.

**Metrics:** We evaluate our model's novel view reconstructions with the same metrics as (Yu et al., 2021): the learned perceptual image patch similarity (LPIPS) metric (Zhang et al., 2018), structural similarity index (SSIM) (Wang et al., 2004), and the peak signal-to-noise ratio (PSNR).

**Results:** We report quantitative comparisons in Tbl. 1 and qualitative comparisons in Fig. 4. Quantitatively, we outperform the baseline (Yu et al., 2021) on all metrics despite being orders of magnitude more efficient. Potential contributing factors for their lower-fidelity reconstructions include an insufficient latent dimensionality imposed by their memory constraints and coarser volumetric sampling yielding less detailed reconstruction.

Table 3: **Memory Consumption and Performance Comparison.** We compare the memory consumption and rendering speed of COLF and uORF in rendering a single image of an encoded scene at 128×128 resolution for various numbers of slots. Our light field decoder yields over an order of magnitude faster rendering and significantly reduced memory overhead. '-' entries indicate results that exceeded available memory.

| | 2 Slots | | 7 Slots | | 60 Slots | |
|---|---|---|---|---|---|---|
| | COLF (ours) | uORF (Yu et al., 2021) | COLF | uORF | COLF | uORF |
| Memory Consumption | 2.4 GB | 7.2 GB | 5.0 GB | 32.0 GB | 31.6 GB | - |
| Rendering Speed | 166 FPS | 6.6 FPS | 50.0 FPS | 1.4 FPS | 6.2 FPS | - |

(a) Scene Editing: Object Manipulation          (b) Scene Editing: Cross-Scene Composition

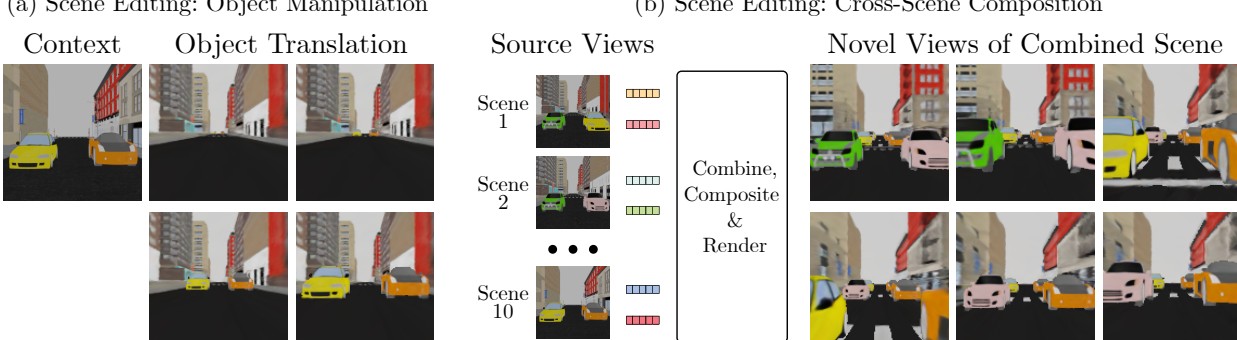

Figure 6: **Editing and rendering of an unbounded scene with many objects.** COLF enables compositing and real-time rendering of unbounded scenes with many objects. (a) We edit the scene by translating two cars along the road. (b) We composite ten different scenes reconstructed from views throughout the city block into a single scene with twenty objects.

### 4.4 Rendering Speed and Memory Consumption

Real-time rendering and memory-efficiency are important properties for downstream applications such as augmented reality. Thus, we compare memory consumption and rendering speed with uORF (Yu et al., 2021) in Tbl. 3 to show how our method yields important gains on those dimensions. The results highlight that employing a volumetric decoder for object-centric scene representations is considerably expensive and renders any potential downstream applications infeasible — even using a modest number of objects pushes the capacity of even large GPUs (seven objects consumes 32GB of memory, e.g.). While uORF (Yu et al., 2021) establishes the direction of combining object-centric representation with 3D-aware representations, an implementation with reasonable support for downstream tasks and applications with larger object sets is important as well. Using the light field as our decoder facilitates such applications, requiring only 5GB of memory to render a scene with seven objects. Even rendering of scenes with up to 60 objects is feasible without any further optimization.

Memory consumption is often not the only concern, but rendering speed as well. Consider a potential application where the rendering speed is of primary concern, such as interactively manipulating objects in virtual reality: iteratively rendering out 100 frames of a scene with four objects at 128x128 resolution takes 1.3 seconds with COLF vs 31.2 seconds with uORF (Yu et al., 2021), making our model uniquely suitable for such applications.

### 4.5 Application: Scene Editing and Composition

We demonstrate editing and composition of an unbounded scene with many objects. The scene features a city block populated with rows of car models, placed randomly throughout the block's length. The large range in depth from the beginning to the end of the block is a challenge for volume rendering-based approaches,

which would require setting near and far planes that enclose the full scene, while also sampling along the ray densely enough to guarantee sufficient sampling of objects along the ray. As seen in Fig. 6, COLF succeeds in decomposing the city-block scene into background and foreground car light fields.

Subsequently, we may edit the scene by transforming the object-centric light field networks and placing additional objects into the scene. Figure 6 visualizes the editing results. By transforming the input coordinates for an object-centric light field, we may translate the corresponding object throughout the scene. By collecting object codes across scenes and compositing them together, we can generate a scene with twenty cars that can still be rendered at interactive frame-rates. Please see the supplemental material for a baseline comparison and further results, including an analysis of the cost of attempting to render these scenes at the same fidelity using prior approaches.

## 5 Discussion

In summary, we have proposed COLF, a novel compositional neural scene representation that parameterizes the 360-degree, 4D light field of a 3D scene by composing it from object-centric neural light fields, thereby enabling the unsupervised discovery of object-centric 3D representations. COLF outperforms previous state-of-the-art approaches in unsupervised scene decomposition, while being two orders of magnitude faster and significantly less memory-intensive. Several exciting directions for future work remain. While COLF improves over the previous state-of-the-art in scene decomposition and compositional novel view synthesis, nevertheless, the scenes we reconstruct are still limited to simple, synthetic scenes. As discussed in the related work, concurrent work improving the performance of the slot attention inference algorithm offers promising directions for extending our object-light-field based representation to more complex 3D scenes. These concurrent works improving the robustness of slot-attention based encoders are rapidly converging on real-world scenes, with a few approaching real-world driving scenes. Yet, equipped with only 2D-based image decoders, these models struggle to reconstruct scenes well and have no explicit 3D understanding. COLF offers a tractable 3D representation to plug-and-play into their architectures. Prior to our contribution, extending these emerging object-centric architectures with 3D representations via the prior SOTA in 3D object representations (Yu et al., 2021) would prove difficult to scale to large real-world datasets due to the expensive cost of volume-rendering. COLF's computational advantage over Yu et al. (2021) would be particularly significant in the application of object-centric learners to real-world driving scenes, where the number of cars in a given scene can be large. With respect to improving COLF's rendering quality, higher-resolution LFNs may be learned by leveraging neural networks with periodic activation functions (Sitzmann et al., 2020) or Fourier Features (Tancik et al., 2020). Incorporating motion into learning and inference may improve the scene decomposition quality and robustness (Kipf et al., 2021). We believe that such improved inference algorithms for compositional scene representations are the next important step toward applying these models to real-world scenes.

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
