# OpenReview forum: "Unsupervised Discovery and Composition of Object Light Fields"
_TMLR — Rejected by TMLR_

### Review · Reviewer_2gPY · 2022-09-25

**Summary Of Contributions:**

The paper introduced a method that uses neural light fields to disentangle objects from the scene in an un-supervised manner. The authors build a parametric, compositive neural light field representation in an efficient manner, and was able to significantly reduce time and memory requirements to discover objects, disentangle objects, and re-render novel views.

**Broader Impact Concerns:**

The work is primarily evaluating on scenes with objects (furnitures, vehicles), and it's primarily applicable to synthetically generated scenes. I see very little ethical concerns for mis-using the work.

**Requested Changes:**

At the current stage, my preliminary rating for this work is borderline. It would be great if authors can provide more discussions to the concerns in the weakness sections above.

**Strengths And Weaknesses:**

[Strengths]
- The paper is addressing an important research problem of disentangling scenes with disjointed objects in an unsupervised manner. I think this is important because it also leverages neural light fields.
- As a proof-of-concept paper, the author presented the work in a relatively clear manner, and the overall presentation and writing is decent.
- Experiments that compared to prior works demonstrate that the proposed COLF algorithm is decent and  it's good to know that the proposed approach is more efficient than uORF.

[Weaknesses]
- The work seems to be a natual extension of [Yu et al., 2021]. While Yu represents even richer information of the scene (such as segmentations), the proposed work COLF only learn to model object using relatively simpler cues (with colors). It's a little un-clear to me why COF is better than uORF. Moreover -- upon checking Fig.4, it's surprising to me that the COLF's rendering is better despite the novel segment is clearly inferior to those of uORF. I wish there are more explainations to the insights of the algorithm.
- The work seems to be building on top of [Yu et al., 2021] + [Sitzmann et al., 2021] + [Locatello et al., 2020], and is primarily using colors as major features to disentangle objects. I get the nature of the proof-of-concept paper, but I would be very interested to see results on cluttered scenes with similar colors.
- In terms of related works, it may be worthwhile to discuss the paper's relationship to traditional layered depth image and its variants in 2D and 3D modelings. Some of the ideas are actually quite related to those.
- Are there any potential applications for this work? It may be an interesting idea to provide some hints.

---

> ### Author Response · Authors · 2022-10-20
> **Author Response**
>
> ### Q1 - uORF provides more rich scene information, particularly segmentations
> This is incorrect: our model provides segmentations as well. The only additional information [Yu et al., 2021] (uORF) provides is rendered depth per their NeRF formulation (whereas we only provide relative ordering of the objects).
>
> ### Q2 - COLF only learns to model objects using simple color cues
> We use the same object encoder as uORF, so we have the same capability to recognize objects. Our difference from uORF is in the 3D representation, where we use light fields instead of radiance fields to significantly improve the training and rendering speed, as well as significantly decrease memory usage. This slot encoder is not just using colors to define objects – see our city-block example where the cars have relatively complex texture and the supplement for scene decomposition on that benchmark. Additionally, many new works improve on the object encoder, using additional temporal supervision or smarter architectures (SAVI, STEVE), and are plug-and-play into our architecture. But please note that the slot encoder is not the focus of this study, but rather our study answers the following question: can we compose light fields to use them as 3D object representations?
>
> ### Q3 - Unclear why COLF is better than uORF, especially as segmentations are worse
> While we actually outperform uORF in most novel view rendering and segmentation metrics, the main reason COLF is “better” than uORF is the drastic memory reduction: uORF, using radiance fields as its 3D representation, can only represent a single-digit number of objects and is extremely memory intensive due to its expensive volume rendering. These limitations prohibit training uORF on scenes with many objects.
> Note that our segmentations are not worse, our model simply converges to assigning the shadows to the object slots instead to the background - this is not wrong, as there is indeed no principled way to decide either way conditioned on the training data, other than hand-crafting a prior. Quantitatively, COLF performs significantly better on the FG-ARI metric which reflects this ambiguity.
>
> ### Q4 - Results on cluttered scenes with similar colors
> Our object encoder and therefore our object-detection capability is the exact same as that of uORF. The object encoder of [Locatello et al.] (Slot attention) does not strictly segment by color; for evidence of this please see the supplement for our decomposition of the city-block scene, where the cars have relatively complex texture. Also please see the supplement of [Yu et al., 2021] which provides generalization evidence to more cluttered scenes using the same encoder. Significant improvements have recently been made to make slot attention robust to real-world scenes (SAVI++,STEVE) that are compatible with our model but not the subject of this paper.
>
> ### Q5 - Relationships to more traditional methods (layered depth images in 2D/3D)
> Indeed there is some similarity to layered depth image representations for 3D modeling and novel view synthesis. However, layered depth images perform small-baseline view synthesis and decompose a scene into depth layers, whereas our representation permits full 360-degree novel view synthesis and decomposes a scene into objects, primarily for the goal of downstream tasks involving object manipulation. We will add a discussion to the related works section.
>
> ### Q6 - Any potential applications for this work, as it is primarily applicable to synthetically generated scenes
> While self-supervised object discovery methods are still largely fundamental research and thus constrained to relatively simple scenes, they have ripe potential applications, especially in robotics applications where object-level representations are useful, but also in generalizable novel view synthesis tasks, as we demonstrate here. Consider a self-driving application where decomposing the scene into the individual cars and representing their location in 3D is a very rich representation for considering object interactions and planning. However, while uORF can only be trained on scenes with few objects and is extremely slow during training and inference, our model is capable of inferring 3D representations with many more cars, which would be critical in the self-driving domain. Our study is how to replace the 3D radiance field representation, which is extremely expensive, with a light-weight 3D representation, the light field. We will extend of possible applications of unsupervised object discovery in the paper.

---

> > ### Author Response · Authors · 2022-11-09
> > **Author response**
> >
> > Dear reviewer 2gPY,
> >
> > We feel that we have adequately addressed the points you raised in your review. Please feel free to raise any further questions with us, and we will be happy to address them.

---

### Review · Reviewer_2uG2 · 2022-10-01

**Summary Of Contributions:**

The paper presents an unsupervised method for compositional modeling of 3D visual scenes.  The proposed method first uses a Slot Attention module to extract a set of object codes from input image and then each code is fed into a variant of Light Field Network, which predicts a color and ordering value given a query ray.  Subsequently, a light field compositor uses an MLP network to compute the ray-visibility for the entire scene and predict the color output for the query ray.  The proposed method is evaluated on four synthetic datasets for scene decomposition, novel view synthesis and scene editing.

**Broader Impact Concerns:**

No concerns on the ethical implication.

**Requested Changes:**

See Weaknesses 2, 3, 5, 6. Please also add more discussion/clarification on 4.

Minor:
- Adding the variance in the segmentation results in Table 2 would make the comparison clearer.
- The FG-ARI values of uORF in Table 2 are different from the published results in Yu et al. ICLR 2022.

**Strengths And Weaknesses:**

Strengths:
1. This work propose to use an alternative NRF representation of objects based on the LFN for unsupervised 3D modeling, which significantly improves the computational efficiency in terms of rendering speed and memory usage.
2. On the synthetic benchmarks, the proposed method achieves strong results on scene decomposition and novel view synthesis.

Weaknesses:
1.  Except for the LFN-based object representations, the entire framework of this work is closely similar to the previous work Yu et al., ICLR2022, including the Slot attention encoder and separate modeling of objects & background.
2. One key component of this work, the ordering value o_i, lacks certain details: 1) What is the value range of o_i? 2) What are learned o's values on the benchmark? It would be clearer if they can be visualized on some examples.
3. The generalization of the learned LFNs. The proposed LFN-based representation encodes the spatial arrangement of objects into ordering prediction, which is scene-specific. It is less clear how such representation, once learned, can be generalized to scenes with very different layouts. It seems challenging for the slot machine encoder to infer such orderings for a novel scene. It would more convincing if the generalization experiments as in Yu et al. can be performed.
4. The quality of scene decomposition seems poor compared to the uORF in Figure 4. The shadow regions induces large segmentation errors. While it seems help the novel view synthesis, such decomposition provides incorrect object boundaries and can have poor generalization. This is indicated by the numerical results in Table 2, the proposed method has much lower ARI values on CLEVR-567.
5. The results on Scene editing are  a bit lacking as no quantitative comparisons with other methods are provided.
6. For the road scene benchmark, it would be more convincing to evaluate the method on some realistic images as many real-world road scene datasets are available.

---

> ### Author Response · Authors · 2022-10-20
> **Author Response**
>
> We thank the reviewer for their constructive feedback. We will post the revised version ASAP.
>
> ### Q1 - Only the light field based object representation is different from Yu et al (uORF)
> Our work observes that uORF succeeds in discovering 3D object-centric representations. However, uORF does not scale to complex scenes such as those with many objects. This limitation stems from the prohibitive cost of volume rendering compositional radiance fields. We focus on addressing this limitation by proposing our COLF representation. This is by no means trivial, as a light field representation requires addressing the question of composing neural light fields that has not been tackled before. We therefore make no improvements to the object discovery module, as that is not the focus of our study and it allows a principled comparison.
>
> ### Q2 - Lacking key details on the ordering values
> The ordering values are raw scalar outputs regressed by the model (unnormalized probabilities that a certain object is first to intersect the ray). They thus approximately align with visibility scores. Note that we visualize the ordering values on the benchmark dataset in the supplement.
>
> ### Q3 - Proposed LFN-based representation is scene-specific and generalization ability is unclear
> This is a misunderstanding: the LFN-based representation is **dynamically inferred in a single forward pass for each scene**, and **not** scene-specific. The results illustrated are on unseen scenes during training, given only a single context image from them at test time (the representations are not optimized per-scene as in the standard NeRF formulation). We use the same object encoder as in uORF, so with respect to the generalization of our object detector, we generalize to the same extent as uORF on scenes with more objects. Also please consider that the focus of our work is not the object encoder, but rather to answer the question of whether we can replace the radiance field of uORF with light fields.
>
> ### Q4 - Quality of scene decomposition seems poor compared to uORF (in shadow regions)
> Please note that there is no principled way for the model to decide whether shadows belong to objects or not - the shadows *are* indeed part of the 3D scene that is causally related to the objects, and, moreover, there is indeed no possible signal in our supervision to assign them to background or the objects, unless we hand-craft an additional prior. Consider recent related work D2NeRF, which similarly performs foreground-background disentanglement, and which similarly assigns the object shadows to the foreground.
>
> Note that our setting is fundamentally different from the *supervised* semantic segmentation setting. In that setting, shadows are typically manually assigned to the background. We agree that this is not obvious and will add a discussion to the results section. Also please recognize that on the FG-ARI metric, which measures only the foreground object segmentations (since the background assignment with shadows is contentious), we actually outperform uORF, so our representation actually disentangles individual objects *better* than uORF.
>
> ### Q5 - Results on scene editing are lacking quantitative comparisons
> Our scene editing demonstration (Figure 5) indeed has no quantitative comparison, but this is because our baseline (uORF) cannot perform in this setting. Please see Figure 1 in the supplement for a demonstration that uORF cannot represent the city-block scene with its large camera range. Though please note that our scene editing looks seamless despite the model having never been queried with any object transformations during training, a generalization we get for free with our object-disentangled formulation.
>
> ### Q6 - Should evaluate on a real-world driving benchmark
> The recent trend of NeRF results has seemingly lifted this field into the real world, and our community is used to seeing results on real data. However, note that that setting is *fundamentally different* than our setting. Currently *no* unsupervised object discovery model has been demonstrated to reliably work on such real-world driving scenes. Recent work (SAVI++) achieves encouraging results, but requires depth signals and object bounding boxes. Our work focuses on the 3D object representation instead of improving the object discovery module, but we are certain advancements in the latter direction will be fruitful in combination with our 3D object representation. uORF is closest to our work, and we outperform uORF in terms of the scenes that we can reconstruct. Real-world results are simply not currently tractable in this line of work.
>
> ### Q7 - FG-ARI Discrepancy with uORF
> We thank the reviewer for catching this discrepancy and will correct this in the revision. Thankfully this only improves our relative results but we apologize for the error.

---

> > ### Author Response · Authors · 2022-11-09
> > **Author response**
> >
> > Dear reviewer 2uG2,
> >
> > We feel that we have adequately addressed the points you raised in your review. Please feel free to raise any further questions with us, and we will be happy to address them.

---

> > ### Comment · Reviewer_2uG2 · 2022-11-22
> > **Overall recommendation**
> >
> > Thanks for the author's feedback and clarification. The author's reply addressed some of my concerns, including the ordering values (Q2), poor object boundary (Q4), and evaluation on the real-world driving scenes (Q6).
> >
> > However, I feel the experimental evaluation on the generalization and scene editing a bit lacking (Q3 & Q5), especially compared with the results and analysis in Table 4 & Section 4.3/4.4 in Yu et al (uORF).
> >
> > Overall, I think this is an interesting paper with good results and a clear focus on the computational efficiency, which seems sufficient for being accepted.

---

### Review · Reviewer_z9ns · 2022-10-13

**Summary Of Contributions:**

The paper tackles the problem of unsupervised object discovery from images. Inspired by recent works in this field that leverage neural rendering (e.g., Nerf) for 3D reasoning, the paper suggests replacing NerF with light field rendering. The main benefit is computational, as rendering an image with light views requires only one evaluation per ray. The paper addressed the challenge that arises from having a light view field per object (instead of a radiance field), which is how to compose the different light views into one rendering. The method is evaluated both qualitative and quantitatively on standard synthetic benchmarks.

**Broader Impact Concerns:**

No broader impact statement has been provided. I do not have any ethical concerns.

**Requested Changes:**

Overall the paper is of good quality. Please see the weaknesses stated above and address them to strengthen the paper.

**Strengths And Weaknesses:**

The paper is well-written and easy to follow. The formulation of the method is clear.
The evaluation seems to be adequate. The scene editing application is nice.
The proposed solution for the composition of light views seems to be simple and reasonable.

A lot of progress has been made in improving radiance fields rendering computation time (e.g., instant ngp). Maybe it is worth mentioning in the computational cost discussion.
I would have liked to see a more thorough discussion regarding the algorithm design choices made in the light view compositor. I think it could also encourage future work. In that regard, consider moving experiments 5.5 and 5.4 from the appendix to the main paper.
Qualtitavly, in terms of object discovery in the main experiment, the results seem to be comparable to nerf.

---

> ### Author Response · Authors · 2022-10-20
> **Author Response**
>
> We thank the reviewer for the careful reading and the constructive feedback! We will incorporate it as follows and post the revised version ASAP.
>
> ### Q1 - Recent advancements in fast NeRF training (e.g. instant ngp) and rendering should be discussed more
> We will add a discussion of recent fast NeRF implementations. We note that these methods often rely on algorithm choices that are *not* compatible with feedforward inference, such as skipping of empty space, non-local hashmaps, or sparse data structures.
>
> ### Q2 - More thorough discussion regarding the design choices for the light field compositor
> We will add a discussion about our design choices for the compositor, and we will move Sections 5.5 and 5.4 to the main paper, space permitting. In particular, we will discuss the following motivation: With the used Plucker parameterization for expressing query rays, which express no camera origin, the only reasonable reference frame for the objects to communicate depth along the query ray is relative to the only common reference point, the context camera position. We considered making the regressed ordering values relative to the projection of the context camera onto the query ray, but because shifting each of the depth values by a constant value preserves the relative depth differences between objects, and because we subtract the minimum of the ordering values as well, making the depths relative to the context camera’s projection point becomes unnecessary.

---

> > ### Author Response · Authors · 2022-11-09
> > **Further questions?**
> >
> > Dear reviewer z9ns,
> >
> > We feel that we have adequately addressed the points you raised in your review. Please feel free to raise any further questions with us, and we will be happy to address them.

---

### Decision · Action_Editors · 2022-12-05

**Recommendation:** Reject

**Comment:**

Overall, the paper received mixed support. The reviewer's concerns about the potential applications, the worse segmentation results with respect to shadows, missing related works, and clarity problems were addressed well. The proposed method is an extension of previous works in unsupervised object discovery via NERF. However, as pointed out by the authors, this extension is not trivial, and does lead to significant computational efficiency. Two reviewers were convinced by this argument, whereas the third was not.

On the negative side, the remaining concerns were:
1) One issue was the missing evaluation of generalization as in Sec 4.4 of Yu et al, to which the authors pointed out that proposed method is not scene-specific and uses the same encoder as the baseline.  The reviewer was not particularly convinced.
2) There are no quantitative evaluation of scene editing (e.g., as in Sec 4.3 of Yu et al). The authors only provided qualitative results on their scene editing task, which also did not convince the reviewer.
3) Results on cluttered scenes with similar colors were also missing. Authors only provided a vague answer about Yu et al also generalizing to more cluttered scenes.

The AC agrees with these three concerns of the reviewers (especially points 1 and 2). As the proposed work extends the baseline uORF (Yu et al), it would be helpful to see the comparison to generalization and scene editing as performed in Yu et al, so as to better understand how the proposed work fits within the context of the previous work. Specifically, while the proposed work is more computationally efficient, does it incur some disadvantages such as generalization ability?  Meanwhile, the scene editing aspect should be quantitatively evaluated in both the scenes in Yu et al, and the large scene in the manuscript. The request for cluttered scenes with similar colors seems to fall under point 1.

The authors are encouraged to revise the paper accordingly for resubmission.

**Audience:**

Neural scene representations are actively researched in the community, and could be applied to robot applications.  The proposed method provides an efficient representation that could be used for these applications.

**Claims And Evidence:**

The major claim is "our method enables unsupervised learning of object-centric neural scene representations, state-of-the-art reconstruction and novel view synthesis performance on standard datasets, and rendering and training speeds at orders of magnitude faster than existing 3D approaches".

This claim is partially supported by evidence via experiments on scene decomposition and novel view synthesis.  However, the paper does not present tests on generalization to cluttered scenes or evaluation of scene editing, as in the baseline work uORF.  Thus, it is unclear if the benefits of the proposed work are  limited to the tested novel view synthesis and computational efficiency.